# Bleeding Events Associated with Rivaroxaban Therapy in Naive Patients with Nonvalvular Atrial Fibrillation: A Longitudinal Study from a Genetic Perspective with INR Follow-Up

**DOI:** 10.3390/medicina60101712

**Published:** 2024-10-18

**Authors:** Nur Ul Ain, Niaz Ali, Abid Ullah, Shakir Ullah, Shujaat Ahmad

**Affiliations:** 1Department of Pharmacology, Institute of Basic Medical Sciences, Khyber Medical University, Peshawar 25100, Khyber Pakhtunkhwa, Pakistan; noorulain.ibms@kmu.edu.pk (N.U.A.); shakir.ibms@kmu.edu.pk (S.U.); 2Institute of Pharmaceutical Sciences, Khyber Medical University, Peshawar 25100, Khyber Pakhtunkhwa, Pakistan; 3Department of Pharmacology, College of Medicine, Shaqra University, Shaqra 11961, Saudi Arabia; 4Department of Pharmacy, Shaheed Benazir Bhutto University Sheringal, Dir Upper 18000, Khyber Pakhtunkhwa, Pakistan; abid@sbbu.edu.pk (A.U.); shujaat@sbbu.edu.pk (S.A.)

**Keywords:** rivaroxaban, pharmacogenetics, ABCB1 gene, CYP3A5 gene, stroke, atrial fibrillation, coagulation parameters, INR

## Abstract

*Background and Objectives*: Rivaroxaban is a direct-acting anticoagulant used to prevent stroke in patients with atrial fibrillation. Rivaroxaban is a substrate for P-glycoprotein, which is encoded by the ABCB1 gene. Rivaroxaban is also metabolized by the CYP3A5 gene. Therefore, the current study is carried out to study the effects of polymorphisms in the ABCB1 and CYP3A5 genes, which may affect the plasma levels of rivaroxaban, with subsequent clinical outcomes (bleeding events) associated with the therapy. *Materials and Methods*: The study was conducted on 66 naive patients with atrial fibrillation treated with rivaroxaban. Blood samples of rivaroxaban were taken at 3 h and after 1 month following the administration of the drug to measure plasma levels. The blood level of rivaroxaban was measured with an HPLC-UV detector. Sanger sequencing was used to find polymorphisms in the targeted genes. Coagulation parameters were measured at 3 h and after 1 month of administration of rivaroxaban. Frequencies of bleeding events were recorded throughout the one-month course of drug therapy. *Results*: The heterozygous and homozygous mutant genotypes of ABCB1 (rs2032582, rs1045642, rs1128503, and rs4148738) and CYP3A5 (rs776746) showed lower plasma concentrations as compared to the wild-type genotype. ABCB1 (rs2032582, rs1045642, rs1128503, and rs4148738) and CYP3A5 (rs776746) gene polymorphisms had a statistically significant impact on the plasma concentration of rivaroxaban among the heterozygous and homozygous mutant genotypes compared to the wild-type genotype. The heterozygous variant of ABCB1 and homozygous variant of CYP3A5 suffered more events of bleeding. *Conclusions*: It was concluded that ABCB1 (rs2032582, rs1045642, rs1128503, and rs4148738) and CYP3A5 (rs776746) gene polymorphisms had a significant impact on the plasma levels of rivaroxaban in patients treated for atrial fibrillation on day three as well as after one month of the therapy. The lowest plasma levels were observed in patients with a homozygous variant of ABCB1 (rs2032582, rs1045642, or rs4148738) along with the CYP3A5*1/*3 allele. The heterozygous variant of ABCB1 SNPs and homozygous variant of CYP3A5 SNPs suffered more events of bleeding.

## 1. Introduction

Atrial fibrillation is a condition in which various regions of the atria rapidly beat faster than the heart’s normal sinus node pacemaker. As a result, the atria only quiver and do not contract properly. This is associated with rapid and erratic atrial activity [1]. As the atrium is not contracting in atrial fibrillation, it results in the stasis of blood, which increases the risk of stroke due to clot formation. A clot is more risky in the left atrial appendage [2]. The primary pharmacologic remedy in atrial fibrillation is to treat the clot and avoid its complications. Treatment includes the use of antiarrhythmic medications to ease symptoms and prevent the stroke associated with atrial fibrillation [3]. The primary oral anticoagulant medication used in healthcare settings is Warfarin. However, it has several disadvantages, which include a gradual onset of action, not enough anticoagulation properties, a low therapeutic index associated with frequent bleeding, and a high frequency of food and drug interactions that require regular laboratory tests and follow-up. In clinical settings, rivaroxaban, apixaban, edoxaban, and dabigatran are now being used as new oral anticoagulants (NOACs). They are distinct in affecting a particular coagulation component (either thrombin or factor Xa) and focus on clot development and the deposition of fibrin [4]. New oral anticoagulants (NOACs) were accepted to decrease the possibility of stroke and systemic embolism in patients having nonvalvular atrial fibrillation (NVAF). This includes venous thromboembolism (VTE) prophylaxis, like in elective orthopedic surgical procedures, and the prevention of clot formation [5]. The benefits of NOACs include a quick start of the action, the easy availability of drugs, a high therapeutic index, no interactions with food, and minimal interactions with drugs. Until now, scientists have suggested that no coagulation tests are required [4]. Rivaroxaban is a direct and specific blocker of activated coagulation factor X (FXa), which is an important enzyme located at the intersection of both intrinsic and extrinsic coagulation processes [6,7]. The oral bioavailability of Rivaroxaban is about 80%. The maximum concentration of the drug reaches the plasma after 2–4 h of administration. Studies suggest that rivaroxaban is 95% bound to plasma proteins. Rivaroxaban is transported by the ABCB1 gene encoded by P-glycoprotein, while the ABCG2 gene is encoded with breast cancer resistance protein (BCRP). On the other hand, Cytochrome P450 isoenzymes 3A5, 3A4, and 2J2 are responsible for the metabolism of two-thirds of the delivered drug, but there may be some other mechanisms of metabolism that do not involve CYP450 [8]. The significant interindividual variation of NOACs may affect clinical outcomes regarding the shape of thromboembolism and/or hemorrhagic events. In addition to other factors, one of the possible reasons for the changeability in pharmacokinetics and pharmacodynamics could be mutations of the genes that are encoded for proteins, responsible for the transport, stimulation, and metabolism of NOACs. These genes include ABCB1, CES1, CYP3A4, ABCG2, and CYP3A5 [9]. These variations influence the pharmacokinetics of many other drugs, which are the substrates of P-glycoprotein [10]. The objectives of this study are to investigate the association between rivaroxaban peak plasma levels and subsequent clinical outcomes in relation to genetic polymorphisms in the ABCB1 and CYP3A5 genes among naive patients with nonvalvular atrial fibrillation in Khyber Pakhtunkhwa, Pakistan. We also targeted checking whether there is an association between genotypes of target genes and the coagulation parameters. By addressing these objectives, this study aims to provide a clearer understanding of the pharmacogenetics of rivaroxaban in this specific population in Khyber Pakhtunkhwa, Pakistan. This will ultimately help the optimal management and treatment of atrial fibrillation patients.

## 2. Results

The patients’ mean age was 70.24 ± 6.04 years. Baseline clinical parameters were in a normal range. The frequencies of allele distribution of the ABCB1 and CYP3A5 genes were consistent with Hardy–Weinberg equilibrium (*p* > 0.05).

### 2.1. Plasma Level of Rivaroxaban and SNPs of ABCB1 and CYP3A5 Genes

The plasma levels for rivaroxaban as CDRmax and CDRss with SNPs are shown in Table 1. For the ABCB1 SNPs, the heterozygous and homozygous mutated groups showed relatively lower plasma concentrations than the wild-type genotype. For ABCB1 (rs1045642 and rs1128503), the CDRss values of rivaroxaban showed statistical significance between the wild (AA) and mutant (AG, GG) genotypes with a *p*-value less than 0.05. For ABCB1 (rs2032582 and rs4148738), the CDRmax level is statistically significant between the wild and mutant genotypes. Furthermore, the CDRss level is statistically significant between the wild and only homozygous mutant genotypes with a *p*-value less than 0.05. In our results, the heterozygous and homozygous mutated groups of CYP3A5*3 displayed lower plasma concentrations than the individuals having wild-type genotypes, as shown in Table 1. Our results revealed a statistically significant association between CYP3A5*3 polymorphism and the CDR values of rivaroxaban, as depicted in Figure 1 and Figure 2. We analyzed the plasma levels of drugs in patients with the CYP3A5*3 gene regarding ABCB1 (rs1045642, rs1128503, rs2032582 and rs4148738) gene variants. The homozygous variants of the ABCB1 gene exhibited lower plasma concentrations in patients simultaneously possessing CYP3A5*1/*3 genotypes, as presented in Table 2 and Table 3.

### 2.2. Effect of ABCB1 Gene Polymorphism on Coagulation Parameters

In our study, coagulation parameters including aPTT, PT, and INR on 3 h and after 1 month of rivaroxaban administration were compared among different genotypes, as shown in Table 4 and Table 5. It was found that PT-INR values were higher for heterozygous and homozygous mutant genotypes as compared to wild genotypes, while aPTT values were lower for heterozygous and homozygous mutant genotypes as compared to wild genotypes. As for coagulation parameters after 3 h, the PT and INR values varied significantly in heterozygous genotypes of rs1128503 having values of *p* < 0.05. The aPTT values after 3 h of administration varied significantly with the heterozygous genotype of rs1045642 and varied significantly with the heterozygous and homozygous genotypes of rs2032582 and rs4148738 with a *p*-value less than 0.05. The PT-INR value after 1 month of rivaroxaban administration varied significantly with the homozygous genotype of rs1045642 compared with the wild-type genotypes.

### 2.3. Clinical Outcomes and Frequency of ADRs and SNPs

#### 2.3.1. Clinical Outcomes/Clinical Success

We observed that during a 1-month follow-up study period, there were no cases of recurrent stroke, thromboembolism, recurrent atrial fibrillation, or major bleeding in the enrolled patients.

#### 2.3.2. Frequency of ADRs

A total of nine (13.6%) minor bleeding incidents occurred in the rivaroxaban study group during follow-up, as presented in Figure 3. There was no fatal or deep visceral hemorrhage. Only epistaxis was observed.

## 3. Discussions

The response of a body to a drug is multifaceted. The impact of genetic variables on drug effects is complicated. Polymorphisms in genetics are sometimes assumed to be one of the primary factors that may lead to interindividual variability. Genetic variations in transporters, metabolic enzymes, and receptors associated with drug delivery cause changes in drug plasma concentration and thereby may affect the efficacy and/or toxicity of the drug in the recipients. ABCB1 and CYP3A5 polymorphisms are expressed differently across ethnic groups, contributing to inconsistent findings. For example, CYP3A5 expressers are more common in African populations compared to Caucasians, which might lead to differing risk profiles in different studies [11,12]. Polymorphisms in the ABCB1 gene that encodes P-glycoprotein (efflux transporter) may cause variations in its expression and activity, which may alter the efficacy and toxicity of drugs. Rivaroxaban is a substrate for P-gp, where polymorphism in the ABCB1 gene may alter the dose–response relationship and subsequent risk of bleeding [13].

We observed a significant association between ABCB1 (rs4148738, rs2032582, rs1045642, and rs1128503) gene mutations concerning the Cmax and at steady-state concentration of rivaroxaban, which explores how genetic variables influence the pharmacokinetics of rivaroxaban and subsequent clinical outcomes.

In our study, we found that for ABCB1-related rs4148738 and rs2032582, there was a statistically significant difference between the wild and mutant genotypes and a decrease in the CDR max values of rivaroxaban, while CDRss was statistically significant between the wild and homozygous mutant genotypes. For rs1045642 and rs1128503 of ABCB1, the decreased CDR max and Css levels of rivaroxaban were statistically significant between the wild (AA) and mutant (AG, GG) genotypes. Our findings suggest that polymorphism in the P-glycoprotein expressed by the ABCB1 gene can affect the peak plasma levels of rivaroxaban.

When studying the association between ABCB1 and CYP3A5 polymorphisms and bleeding events in patients using anticoagulants, some findings align with existing studies, while others diverge. This variance can arise due to differences in study design, population demographics, and the drugs being studied. Several studies have shown an association between ABCB1 3435C > T polymorphisms and increased bleeding risk. For example, a study reported by Shnayder et al. [14] found that patients with the TT genotype (associated with reduced P-gp expression) experienced higher plasma concentrations of DOACs, leading to an increased bleeding risk. Similar results were seen in a study reported by Lund et al. [15], where dabigatran-treated patients with the 3435C > T polymorphism had a greater risk of bleeding. Not all studies confirm the link between ABCB1 variants and increased bleeding risk. Some studies report no significant association between the 3435C > T variant and clinical outcomes. For example, research conducted by Cosmi et al. [16] found no significant differences in bleeding risk among different ABCB1 genotypes in patients taking rivaroxaban. One potential explanation for these divergent findings is that drug-specific effects play a role: P-gp expression may impact some DOACs more than others due to varying degrees of reliance on this transporter. Differences in study populations may also explain the inconsistencies. Studies conducted in Asian populations (e.g., reported by Eichelbaum et al. [17] found that the frequency of the 3435T allele was lower, which may affect the ability to detect a significant association. 

Similar types of reports exist that describe the role of ABCB1. Wu et al. identified a significant relationship between the ABCB1 gene (rs4148738) polymorphism and the trough level of rivaroxaban in patients experiencing nonvalvular atrial fibrillation [18]. Wang, Chen, et al. revealed a significant association between the wild and mutant genotypes of the ABCB1 (rs1128503) gene and the concentration of rivaroxaban [13]. These results are consistent with our findings in several respects. Overall, there is a dearth of research available on the impact of gene variations on the clinical implications and pharmacokinetics of rivaroxaban. Some reports discovered no statistically significant relevance between the ABCB1 (rs2032582, rs1045642, rs1128503, and rs4148738) mutations and inter-individual changes in the pharmacokinetics of rivaroxaban [13,19,20,21].

The CYP3A5*3 gene polymorphism is linked to reduced enzyme function, but interestingly, there are also contradictory findings that show a higher elimination of the drug in subjects with the CYP3A5*3 allele as compared to those with the normal allele (CYP3A5*1) [22]. As the CYP3A5*3 variant is a poor metabolizer and plasma levels were decreased in these patients, therefore, the role of CYP3A5*3 is not predominant. Our study explores the significant association of CYP3A5*3 genetic variants on the pharmacokinetics of rivaroxaban. Our research findings are different than the findings of Nakagawa, Kinjo et al., which did not find any significant differences in the trough/D of rivaroxaban amongst CYP3A5*3 (rs776746) individuals [20]. Similarly, Wu, Wu et al.’s study on NVAF discovered no significant change between the wild-type and CYP3A5 (rs776746) mutant genotype and rivaroxaban therapy [18]. Sychev, Minnigulov et al. discovered no statistically significant change in rivaroxaban plasma concentration among mutant and wild genotypes of CYP3A5 (rs776746) [19].

In the perspective of coagulation parameters, Da, Rm et al. stated that rs1045642 polymorphism has a significant effect on changing the PT values in atrial fibrillation patients treated with rivaroxaban [23]. However, once we plotted the CYP3A5*3 versus the ABCB1 gene, it is predicted that the homozygous variants of the ABCB1 (rs2032582, rs1045642, and rs4148738) gene suffered the lowest plasma levels in the patients that simultaneously contained the CYP3A5*1/*3 allele. Therefore, homozygous variants of the ABCB1 gene should be screened for the CYP3A5*3 gene to avoid therapy failure. The Nakayama et al. [24] and Sheikh et al. [25] study findings are similar to our observations regarding minor bleeding.

According to a case report, an elderly patient having atrial fibrillation experienced bleeding that resulted from ABCB1 gene polymorphism [7]. From the perspective of coagulation parameters, there are reports of bleeding, which are presented in Figure 3. CYP3A5 is a key enzyme involved in the metabolism of several anticoagulants, particularly DOACs like rivaroxaban and apixaban. Patients who carry loss-of-function polymorphisms, such as the **CYP3A53/3 genotype, exhibit reduced enzyme activity, leading to decreased drug clearance and higher plasma concentrations of the anticoagulant [26].

ABCB1 (P-glycoprotein) plays a crucial role in the transport of anticoagulants across membranes, affecting their absorption, distribution, and excretion. Variants like ABCB1 3435T have been shown to reduce P-glycoprotein expression, potentially increasing drug exposure and the risk of bleeding. This is particularly relevant for DOACs such as dabigatran, which are substrates of P-glycoprotein [27]. A study by Wang et al. [13] examined the impact of ABCB1 polymorphisms (e.g., rs1045642) on rivaroxaban pharmacokinetics in a cohort of European patients. The researchers found that carriers of the *C* allele had significantly lower plasma concentrations of rivaroxaban, which correlated with an increased risk of thromboembolic events compared to non-carriers. This underscores the need for genetic screening in European populations to tailor rivaroxaban dosing [13]. In contrast, Wu et al. [18] studied the association of ABCB1 polymorphisms in a cohort of East Asian patients. They reported that the prevalence of the *C* allele was lower in this population, leading to different dosing requirements and therapeutic outcomes. This study highlighted the importance of population-specific genetic variations and their influence on anticoagulant therapy. Research conducted by Kryukov et al. [28] evaluated the impact of the CYP3A5*3 and CYP3A5*6 alleles on rivaroxaban pharmacokinetics among European patients. The study found that individuals with the CYP3A5 expressers genotype (e.g., CYP3A51/*1) had significantly higher rivaroxaban clearance, leading to lower plasma concentrations and a reduced risk of bleeding compared to non-expressers. This variability necessitates careful consideration of genetic testing in this population to optimize dosing strategies. A study by Li et al. [29] focused on East Asian populations and reported a higher prevalence of expressers compared to European cohorts. The findings suggested that the dosing regimens currently recommended might not be sufficient for these patients, as expressers may require dose adjustments to achieve therapeutic levels without increasing bleeding risk. This highlights the need for region-specific dosing guidelines. The beauty of our study is that it describes the importance of heterozygous variants of ABCB1 SNPs, as it observed more frequencies of bleeding as compared to the reference or homozygous variants. The shift in the INR values was also significant. As PT and INR are related, all patients’ PT and aPTT values are prolonged, suggesting improvement in this therapy. However, in ABCB1 heterozygous variants, there was more frequencies of epistaxis as compared to other reference or homozygous variants.

## 4. Materials and Methods

### 4.1. Participant Recruitment

A total of 66 naive patients with nonvalvular atrial fibrillation were enrolled in this study. The study was carried out at Khyber Teaching Hospital and Hayatabad Medical Complex, Peshawar, from August 2022 to February 2024. The patients were enrolled using a written consent form in the local language. The patients received rivaroxaban 20 mg, 15 mg, and 10 mg with a once-a-day schedule as instructed by the respective physicians according to clinical guidelines and the clinical status of the patients. The Advanced Study and Research Board of Khyber Medical University, Peshawar, approved the study protocols via no. DIR/KMU-AS&RB/AC/001794. The Ethical Board of Khyber Medical University accorded approval via no. DIR/KMU/IBMS/IRBE/meeting/2022/9303-5 dated 12 October 2022. The sample size (n = 66) was calculated using OpenEpi (https://www.openepi.com/SampleSize/SSCohort.htm, accessed on 28 September 2024). Patients were chosen based on the following criteria: 80% power (1-beta, % chance of detection), 95% two-sided significance level (1-alpha), and the ratio of sample size (unexposed/exposed is 1, percent of unexposed outcome is 25%, and percent of exposed with the outcome is 50%).

On a standardized (in the local language) informed consent, the participants were informed about the aims and purpose of the study. They were told that the study aims to investigate the association between ABCB1/CYP3A5 polymorphisms and bleeding events in patients using anticoagulants. The genetic aspects of the study were also explained to them, such as the analysis of their DNA to detect specific polymorphisms. All these explanations were in the local simple and non-technical language.

#### 4.1.1. Inclusion Criteria 

Naive patients were enrolled on the following eligibility conditions: (1) age greater than 65 years; (2) verification of confirmed cases of NVAF and receiving anticoagulant treatment; (3) willingness to give written informed consent for participation. 

#### 4.1.2. Exclusion Criteria

The main criteria for exclusion were as follows: (1) confirmed diagnosis of valvular AF in which there is the presence of artificial heart valves and mitral stenosis; (2) DOACs in combination with CYP3A4/5 or P-gp inducers or inhibitors; (3) co-morbidities with a higher risk of bleeding such as thrombocytopenia and peptic ulcer; (4) inability to attend the planned follow-up visits [13].

### 4.2. Study Design

This is a prospective two-center hospital-based longitudinal study. The patients’ demographic and clinical parameters such as gender, age, serum creatinine, alanine aminotransferase, aspartate aminotransferase, troponin, blood urea, and lipid profile tests were recorded at the start of the study. The PT, INR, and aPTT tests were performed to assess the clinical responses at baseline before the administration of rivaroxaban, after 3 h of the drug administration, and after 1 month with continuous administration of the drug. Clinical outcomes and the frequency of major and minor bleeding events were recorded. Major bleeding was defined as bleeding that resulted in a hemoglobin loss of at least 20 g/L, transfusion of at least 2 units of packed red blood cells, bleeding into a closed compartment (such as the retroperitoneum or cranium), or bleeding that caused death [30]. Minor bleeding events, on the other hand, were characterized as acute clinically noticeable events that did not fit the criteria for major bleeding [31].

### 4.3. Analysis of Plasma Concentrations

High-performance liquid chromatography-UV detection was used to assess the plasma levels of rivaroxaban using a C18 column [32,33]. To determine the peak plasma concentration (Cmax) of rivaroxaban, blood samples were collected 3–4 h (Tmax) after the administration of the drug. Plasma levels were also collected at one-month follow-up to correlate for clinical outcomes and possible shifts in the plasma levels of the recipients. Briefly, blood samples were centrifuged for 15 min at 3000 rpm to extract plasma. Eppendorf tubes were used for the collection of segregated plasma and refrigerated at −70 °C for further analysis. Under isocratic conditions, analytes were eluted with a mobile phase A/mobile phase B ratio of 90:10 *v*/*v*. Mobile phase A was acetonitrile and mobile phase B consisted of water. The samples were deproteinized with acetonitrile. Amounts of 200 µL of plasma, 200 µL of acetonitrile, and 20 µL of internal standard were mixed. The mixture was agitated for 30 s and centrifuged for 5 min at 16,000 rpm to separate proteins. The supernatant was collected in a clean Eppendorf tube and once again centrifuged for 5 min at maximum speed. The extract was filtered by using a Nylon membrane filter micropore with a pore size of 0.45 µm. A reconstituted sample of about 20 µL was introduced into the injection port of the HPLC equipment, where the mobile-phase flow rate was adjusted to 1 mL/min with a retention time of 10 min. Detection of rivaroxaban was carried out at 249 nm. The protocol was repeated through inter-day and intra-day assay. The coefficient of variance, accuracy, precision, limit of detection, and percent yield were determined to validate the method. The limit of detection was determined using a serial dilution of the standard drug through spiking.

### 4.4. Extraction of Genomic DNA

Genomic DNA was extracted from whole blood using a Wiz Prep™ gDNA Mini Kit (Wiz BioSolutions, Seongnam, Republic of Korea). The Nanodrop spectrophotometry method and gel electrophoresis were used to confirm the purity and concentration of isolated DNA. Polymorphisms in the ABCB1 gene at sites with rs1045642, rs1128503, rs4148738, and rs2032582 and for the CYP3A5 (rs776746) gene were investigated. The amplification of targeted genes was performed using a gradient thermocycler. Primers were constructed with the help of the UCSC genome database and relevant literature [34,35]. The PCR results were run on a 2% Agarose gel [36].

### 4.5. Sanger Sequencing

The Sanger sequencing method was used to determine the sequence of targeted sites of the ABCB1 and CYP3A5 genes using a Seq Studio TM machine (Applied biosystems, Foster City, CA, USA). The amplification of the sequenced product was conducted using the specifications provided by the manufacturer. A Big Dye X-terminator TM kit (Thermo Fischer Scientific, Norristown, PA, USA) was used to optimize the sequenced compounds. The analysis was completed when the samples were loaded, and after a medium run-on, electrophoresis was established thereafter. The analysis of genotypes distribution was performed on the global population using a Hardy–Weinberg online calculator (Hardy Corporation, Birmingham, AL, USA). A Finch TV supplied with version 1.4 software was used to investigate the sequences for the presence of single-nucleotide mutations [37,38].

### 4.6. Statistical Analysis

The rivaroxaban plasma levels, and coagulation parameters were represented as mean ± SD. Genotypes and bleeding events were presented in frequencies and percentages. The distribution of genotypes and allele frequencies was tested to comply with Hardy–Weinberg equilibrium to compare with world genotype distributions. A one-way ANOVA followed by Tukey’s post hoc test was applied for multiple comparisons among different groups of rivaroxaban with all genotypes of the ABCB1 and CYP3A5 genes. A one-way ANOVA followed by Tukey’s post hoc test was applied to compare the coagulation tests among the different genotypes of targeted genes. Bleeding events were compared among all genotypes using Fisher’s exact test followed by Yat’s correction. SPSS version 22 was used to evaluate the data, and Microsoft Excel 365 was used for graph making. The results are presented in figures and tables.

## 5. Conclusions

We found a significant impact of ABCB1 (rs4148738, rs1045642, rs1128503, and rs2032582) gene mutations on the peak plasma levels in rivaroxaban therapy. For ABCB1 (rs2032582, rs4148738, and rs1045642), heterozygous variants suffered more bleeding events, with a statistically significant shift in PT-INR values between the wild and heterozygous mutant genotypes of ABCB1 (rs1128503). We anticipate that the results of our study will be helpful for the future investigation of the impact of genetic factors on the pharmacokinetics and clinical outcomes of rivaroxaban from the perspectives of INR values.

### Limitations of the Study

We conducted our longitudinal study on 66 naive patients. We recommend similar types of studies in a higher number of patients in target populations before extrapolating the results of this study.

## Figures and Tables

**Figure 1 medicina-60-01712-f001:**
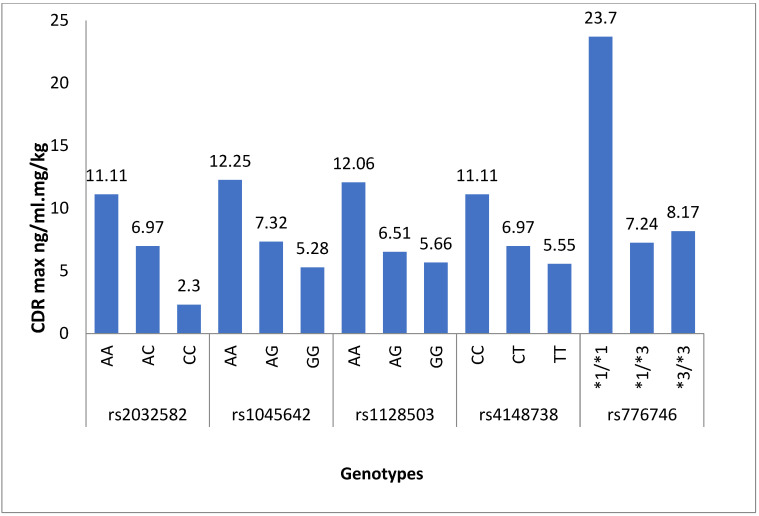
Comparison of rivaroxaban max plasma levels among genotypes at 3 h of rivaroxaban administration.

**Figure 2 medicina-60-01712-f002:**
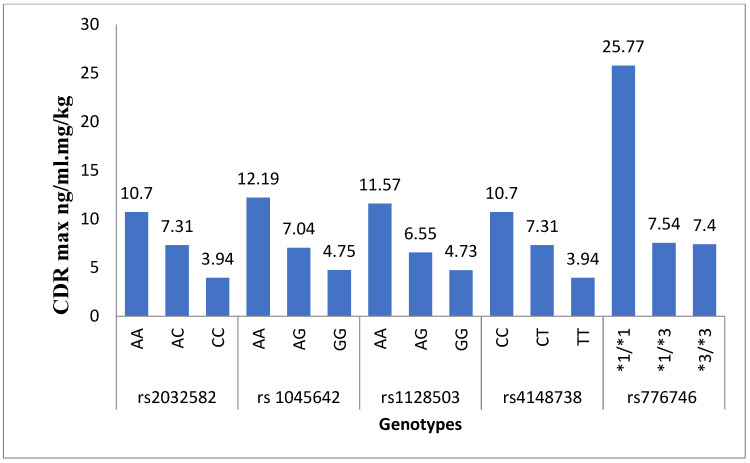
Comparison of rivaroxaban steady-state plasma levels among genotypes at 1 month of rivaroxaban administration.

**Figure 3 medicina-60-01712-f003:**
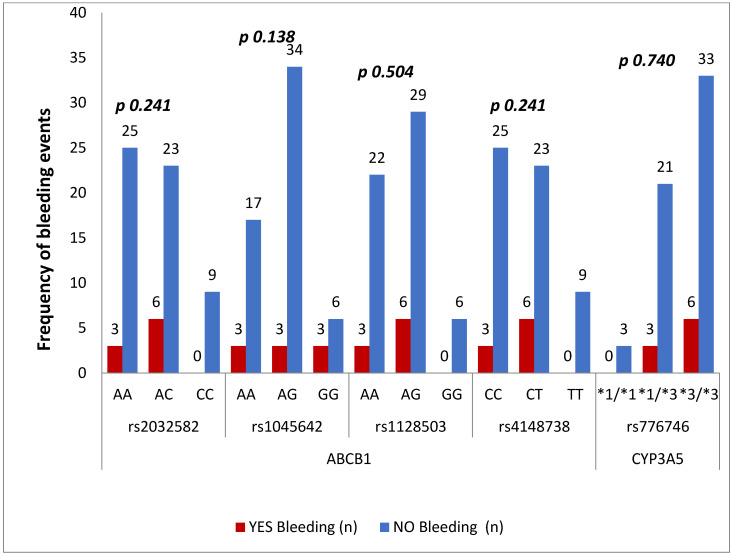
The bleeding events of different genotypes.

**Table 1 medicina-60-01712-t001:** Comparison of rivaroxaban plasma level among genotypes (Data are presented as mean ± SD).

GENES	SNPs	Genotypes	CDR Max (3 h)	*p*-Values	CDR Steady State (1 Month)	*p*-Values
ABCB1	rs1128503	AA	12.06 ± 7.31	-	11.57 ± 8.24	-
		AG	6.51 ± 4.15	0.001 *	6.55 ± 3.56	0.004 *
		GG	5.66 ± 2.42	0.034 *	4.73 ± 0.74	0.029 *
	rs1045642	AA	12.25 ± 8.24	-	12.19 ± 9.05	-
		AG	7.32 ± 4.17	0.007 *	7.04 ± 3.62	0.005 *
		GG	5.28± 2.69	0.008 *	4.75 ± 1.19	0.005 *
	rs2032582	AA	11.11 ± 7.43	-	10.70 ± 8.18	-
		AC	6.97 ± 4.35	0.022 *	7.31 ± 3.48	0.081
		CC	2.30 ± 0.76	0.036 *	3.94 ± 1.10	0.010 *
	rs4148738	CC	11.11 ± 7.43	-	10.70 ± 8.18	-
		CT	6.97 ± 4.35	0.022 *	7.31 ± 3.48	0.081
		TT	5.55 ± 2.30	0.036 *	3.94 ± 1.10	0.010 *
CYP3A5	rs776746	*1/*1	23.70 ± 4.32	-	25.77 ± 3.31	-
		*1/*3	7.24 ± 6.16	0.001 *	7.54 ± 6.37	0.001 *
		*3/*3	8.17 ± 4.57	0.001 *	7.40 ± 4.05	0.001 *

One-way ANOVA followed by Tukey’s post hoc test was used; * = statistical significance (*p* < 0.05). - shows reference values.

**Table 2 medicina-60-01712-t002:** Rivaroxaban plasma levels in the context of CYP3A5 verses ABCB1 genotypes after 3 h of drug administration.

CYP3A5 rs776746	ABCB1											
	rs2032582			rs1045642			rs1128503			rs4148738		
	AA	AC	CC	AA	AG	GG	AA	AG	GG	CC	CT	TT
*1/*1	237.03 ± 0.12	-	-	237.03 ± 0.12	-	-	237.03 ± 0.12	-	-	237.03 ± 0.12	-	-
*1/*3	132.79 ± 62.39	121.27 ± 85.96	53.00 ± 0.14	137.56 ± 79.81	127.28 ± 66.52	36.48 ± 0.14	155.49 ± 55.08	96.30 ± 72.38	-	132.79 ± 62.39232	121.27 ± 85.96	53.00 ± 0.13
*3/*3	167.64 ± 86.83	110.44 ± 39.66	120.50 ± 2.63	147.40 ± 78.24	125.75 ± 62.29	36.48 ± 0.14	167.64 ± 86.83	118.52 ± 35.46	93.54 ± 26.89	167.64 ± 86.83	110.44 ± 39.66	120.50 ± 2.63

Data are presented as mean ± SD; - means this group contains no patients.

**Table 3 medicina-60-01712-t003:** Rivaroxaban plasma levels in the context of CYP3A5 versus ABCB1 genotypes after 1 month of drug administration.

CYP3A5 rs776746	ABCB1											
	rs2032582			rs1045642			rs1128503			rs4148738		
	AA	AC	CC	AA	AG	GG	AA	AG	GG	CC	CT	TT
*1/*1	257.75 ± 0.23	-	-	257.75 ± 0.23	-	-	257.75 ± 0.23	-	-	257.7588 ± 0.23	-	-
*1/*3	124.26 ± 64.36	139.15 ± 63.43	64.74 ± 0.21	148.86 ± 87.48	113.60 ± 57.59	113.49 ± 0.21	142.68 ± 64.57	110.23 ± 60.37	-	124.26 ± 64.36	139.15 ± 63.43	64.74 ± 0.25
*3/*3	167.64 ± 86.83	119.19 ± 57.65	72.35 ± 9.70	128.73 ± 69.11	131.89 ± 68.52	113.49 ± 0.21	148.75 ± 79.80	116.57 ± 59.80	81.10 ± 0.12	148.75 ± 79.80	119.19 ± 57.65	72.35 ± 9.70

Data are presented as mean ± SD; - means this group contains no patients.

**Table 4 medicina-60-01712-t004:** Impact of ABCB1 and CYP3A5 polymorphisms on coagulation parameters at 3 h after rivaroxaban administration.

GENES	SNPs	Genotypes	PT at 3 h (s) (11–13.5 s)	*p*-Values	INR at 3 h (2–3)	*p*-Values	APTT at 3 h (s) (26–34 s)	*p*-Values
ABCB1	rs1128503	AA	18.00 ± 2.57	-	1.49 ± 0.24	-	34.78 ± 6.48	-
		AG	23.50 ± 9.61	0.015 *	2.05 ± 0.97	0.013 *	34.24 ± 10.46	0.972
		GG	16.50 ± 3.28	0.894	1.38 ± 0.26	0.941	33.00 ± 7.66	0.900
	rs1045642	AA	19.41 ± 4.20	-	1.60 ± 0.36	-	40.25 ± 11.32	-
		AG	20.85 ± 9.72	0.782	1.81 ± 0.98	0.593	31.30 ± 6.09	0.001 *
		GG	23.53 ± 3.09	0.389	2.06 ± 0.33	0.302	33.63 ± 5.26	0.104
	rs2032582	AA	18.96 ± 3.73	-	1.57 ± 0.32	-	37.66 ± 10.45	-
		AC	22.29 ± 10.82	0.242	1.95 ± 1.08	0.152	32.46 ± 4.91	0.059 *
		CC	21.56 ± 3.55	0.655	1.85 ± 0.38	0.607	30.00 ± 10.21	0.053 *
	rs4148738	CC	18.96 ± 3.73	-	1.57 ± 0.32	-	37.66 ± 10.45	-
		CT	22.29 ± 10.82	0.242	1.95 ± 1.08	0.152	32.46 ± 4.91	0.059 *
		TT	21.56 ± 3.55	0.655	1.85 ± 0.38	0.607	30.00 ± 10.21	0.053 *
CYP3A5	rs776746	*1/*1	15.00 ± 3.21	-	1.20 ± 1.20	-	33.00 ± 4.31	-
		*1/*3	24.53 ± 10.87	0.090	2.15 ± 1.10	0.095	33.31 ± 12.29	0.998
		*3/*3	18.91 ± 4.06	0.644	1.59 ± 0.39	0.637	35.06 ± 6.31	0.920

Values are expressed as the mean and standard deviation (SD). One-way ANOVA followed by Tukey’s post hoc test was used to compare the coagulation tests among different genotypes.* Represents significant difference (*p* < 0.05). - shows reference values.

**Table 5 medicina-60-01712-t005:** Impact of SNP ABCB1and CYP3A5 polymorphisms on coagulation parameters at 1 month after rivaroxaban administration.

GENES	SNPs	Genotypes	PT at 1 Month (s) (11–13.5 s)	*p*-Values	INR at 1 Month (2–3)	*p*-Values	aPTT at 1 Month (s) (26–34 s)	*p*-Values
ABCB1	rs1128503	AA	16.92 ± 3.21	-	1.42 ± 0.30	-	35.34 ± 4.68	-
		AG	18.94 ± 7.65	0.405	1.62 ± 0.72	0.385	36.84 ± 8.42	0.681
		GG	14.30 ± 1.42	0.602	1.21 ± 0.12	0.690	31.20 ± 0.21	0.383
	rs1045642	AA	16.77 ± 2.54	-	1.40 ± 0.25	-	35.47 ± 5.13	-
		AG	16.44 ± 5.02	0.972	1.39 ± 0.48	0.998	35.37 ± 8.05	0.999
		GG	25.33 ± 9.73	0.001 *	2.20 ± 0.91	0.001 *	38.00 ± 5.26	0.640
	rs2032582	AA	16.80 ± 3.05	-	1.41 ± 0.29	-	34.66 ± 4.84	-
		AC	18.55 ± 7.69	0.525	1.60 ± 0.74	0.421	36.53 ± 8.98	0.569
		CC	18.13 ± 7.40	0.836	1.50 ± 0.59	0.909	36.66 ± 4.44	0.734
	rs4148738	CC	16.80 ± 3.05	-	1.41 ± 0.29	-	34.66 ± 4.84	-
		CT	18.55 ± 7.69	0.525	1.60 ± 0.74	0.421	36.53 ± 8.98	0.569
		TT	18.13 ± 7.40	0.836	1.50 ± 0.59	0.909	36.66 ± 4.44	0.734
CYP3A5	rs776746	*1/*1	14.50 ± 1.21	-	1.20 ± 1.02	-	32.00 ± 2.21	-
		*1/*3	18.33 ± 7.05	0.563	1.57 ± 0.67	0.555	33.81 ± 4.68	0.900
		*3/*3	17.64 ± 5.63	0.668	1.49 ± 0.51	0.684	37.24 ± 7.92	0.405

Values are expressed as the mean and standard deviation (SD). One-way ANOVA followed by Tukey’s post hoc test was used to compare the coagulation tests among different genotypes; * shows statistical significance at *p* < 0.05. - shows reference values.

## Data Availability

The datasets for the current study are available from the corresponding authors on reasonable request.

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
