# Peer review of "Bleeding Events Associated with Rivaroxaban Therapy in Naive Patients with Nonvalvular Atrial Fibrillation: A Longitudinal Study from a Genetic Perspective with INR Follow-Up"

_medicina, 2024, doi:10.3390/medicina60101712_

Round 1

Reviewer 1 Report

Comments and Suggestions for Authors

SUMMARY

The  document appears to be a research article titled "Bleeding events associated with Rivaroxaban therapy in naive patients with nonvalvular atrial fibrillation: a longitudinal study in genetic perspective with INR follow up." It focuses on the pharmacogenetics of Rivaroxaban, specifically looking at polymorphisms in the ABCB1 and CYP3A5 genes, their effect on plasma concentrations, and the subsequent impact on bleeding events and coagulation parameters in patients. The study involved 66 patients, measured plasma levels, and assessed coagulation parameters such as PT, INR, and aPTT over a one-month period.

Please make the following revisions to the document:

Major Revisions:

1. Study Design Clarifications: Provide detailed information on patient recruitment, including inclusion and exclusion criteria. Address potential biases in patient selection.

2. Statistical Analysis Details: Enhance the section on statistical analysis. Explain the choice of tests, especially for comparing different genotypes. Discuss how multiple comparisons were corrected, if applicable.

3. Genotype and Bleeding Association: Further explain the association between ABCB1/CYP3A5 genotypes and bleeding events. Discuss potential mechanisms and integrate a discussion on how these results align or conflict with existing literature.

4. Discussion Section Expansion: Provide a comprehensive comparison with previous studies. Analyze why some findings differ from existing studies and discuss the implications for clinical practice.

5. Study Limitations: Explicitly outline the study's limitations, including the small sample size and potential confounding variables. Discuss how these limitations might influence the results and conclusions drawn from the study.

6. Data Presentation Improvement: Ensure that tables and figures are clear, include error bars in figures showing plasma concentration levels, and label statistically significant differences more prominently. Ensure consistent formatting and sufficient detail in all tables.

7. Ethical Considerations and Consent: Elaborate more on the informed consent process and measures taken to ensure patient safety.

8. Clinical Relevance: Strengthen the discussion on the clinical implications of the findings and how genetic insights could influence current anticoagulation management practices for atrial fibrillation patients.

9. Clarify Objectives: Present the objectives more clearly and specify what you aim to prove regarding the impact of genetic polymorphisms on rivaroxaban plasma levels and clinical outcomes.

10. Additional Analysis: If possible, conduct additional subgroup analyses to determine if specific patient groups are more affected by these genetic variants.

Minor Revisions:

1. Abstract Clarity: Improve the clarity of the abstract by summarizing key findings and their implications more concisely.

2. Terminology Consistency: Ensure consistent use of terms throughout the paper for enhanced readability.

3. Grammar and Language: Proofread the manuscript for grammatical errors, awkward phrasing, and typos.

4. Reference Formatting: Correct any inconsistencies in reference formatting to ensure uniform presentation.

5. Abbreviations: Define all abbreviations at their first mention in the text and add a list of abbreviations for easy reference.

6. Conclusions: Reframe the conclusions to align more closely with the presented data and emphasize the relevance and limitations of the results.

7. Formatting Issues: Correct minor formatting issues, such as inconsistent spacing, font size, or alignment in tables and figures.

Author Response

Point by point comments Answers

Author's Reply to the Review Report (Reviewer 1)

Comments and Suggestions for Authors

SUMMARY The document appears to be a research article titled "Bleeding events associated with Rivaroxaban therapy in naive patients with nonvalvular atrial fibrillation: a longitudinal study in genetic perspective with INR follow up." It focuses on the pharmacogenetics of Rivaroxaban, specifically looking at polymorphisms in the ABCB1 and CYP3A5 genes, their effect on plasma concentrations, and the subsequent impact on bleeding events and coagulation parameters in patients. The study involved 66 patients, measured plasma levels, and assessed coagulation parameters such as PT, INR, and aPTT over a one-month period.

Please make the following revisions to the document:

Major Revisions:

  1. 1. Study Design Clarifications: Provide detailed information on patient recruitment, including inclusion and exclusion criteria. Address potential biases in patient selection.
  • Answer: The biases were controlled through inclusion and exclusion criteria.

Inclusion criteria:

  • Newly diagnosed non-valvular atrial fibrillation patients, who were willing to participate in the study on inform consent.
  • Men or women aged ≥60 years with non-valvular atrial fibrillation
  • Atrial fibrillation must be documented by electrocardiogram (ECG) evidence in order to participate in the optional pharmacogenomic component, subjects must have signed the informed consent for DNA research document indicating willingness to participate in the pharmacogenomics component of the study

Exclusion criteria:

  • Subjects having Cardiac associated conditions like mitral valve stenosis, Prosthetic heart valve, Valvular Atrial fibrillation and Hemorrhage associated conditions like history of intraocular, intracranial, spinal, or intra-articular atraumatic bleeding were not be the part of our study.
  • Subjects who received CYP3A4/5, P-glycoprotein inhibitors and CYP3A4/5, P-glycoprotein inducers were also excluded from the study.
  • Subjects who are contraindicated to Apixaban and Rivaroxaban were excluded from the study.
  1. Statistical Analysis Details: Enhance the section on statistical analysis. Explain the choice of tests, especially comparing different genotypes. Discuss how multiple comparisons were corrected, if applicable.
  • Answer:

The section is enhanced by further elaborating the statistical analysis. We preferred ANOVA for multiple genotypes followed by post hoc correction (Tukey’s test). All corrections are done in view of following statement:

In case if the data follows a normal distribution (parametric data), tests such as ANOVA (Analysis of Variance) were commonly used when comparing more than two groups. For multiple genotypes or groups, ANOVA is preferred when data is normally distributed. If a significant difference is detected, a post-hoc test (e.g., Tukey's HSD) is applied to pinpoint specific differences between the groups. When multiple comparisons are made, there is an increased risk of obtaining false positives (Type I errors). To control this, correction methods are employed: After an ANOVA, Tukey's HSD can be applied for pairwise comparisons between group means, adjusting for multiple comparisons. This method assumes homogeneity of variances and works well for normally distributed data.

So necessary changes are done.

  1. Genotype and Bleeding Association: Further explain the association between ABCB1/CYP3A5 genotypes and bleeding events. Discuss potential mechanisms and integrate a discussion on how these results align or conflict with existing literature.
  • Answer:

We enhanced the literature as advised  in the discussion section.

Briefly stating, reduced P-gp (due to ABCB1 variants) and decreased metabolism (due to CYP3A5 variants) can both lead to increased drug concentrations in plasma, thereby heightening the risk of over-anticoagulation and subsequent bleeding. Variations in the ABCB1 gene, particularly the 3435C>T polymorphism, have been studied for their impact on drug disposition. The T allele is often associated with reduced P-gp expression or activity, leading to altered pharmacokinetics.

  1. Discussion Section Expansion: Provide a comprehensive comparison with previous studies. Analyze why some findings differ from existing studies and discuss the implications for clinical practice.
  • Answer:

As asked, comprehensive discussion is added to previous literature.

The role of ABCB1 and CYP3A5 polymorphisms in bleeding events is complex and influenced by various factors, including population genetics, the pharmacokinetics of different anticoagulants, and clinical factors. While many studies align in showing that these polymorphisms can increase the risk of bleeding, the findings are not universally consistent, highlighting the need for further research and tailored clinical approaches.

The beauty of our study is that we studied the CYP isoforms with ABCB1 genotypes of interest simultaneously to reach a conclusion.

  1. Study Limitations: Explicitly outline the study's limitations, including the small sample size and potential confounding variables. Discuss how these limitations might influence the results and conclusions drawn from the study.
  • Answer:

We added a study limitations section. We are thankful to the reviewer for his/ her guidance.

One of the key limitations in many pharmacogenetic studies, including those investigating ABCB1 and CYP3A5 polymorphisms and bleeding events, is the small sample size. With a limited number of participants, the statistical power of the study is reduced, making it more difficult to detect significant associations or generalize the findings to the broader population.

  1. Data Presentation Improvement: Ensure that tables and figures are clear, include error bars in figures showing plasma concentration levels, and label statistically significant differences more prominently. Ensure consistent formatting and sufficient detail in all tables.
  • Answer: Done accordingly
  1. Ethical Considerations and Consent: Elaborate more on the informed consent process and measures taken to ensure patient safety.
  • Answer: As the informed consent process begins with clearly communicating the study’s purpose to potential participants. In this case, participants were informed that the study aims to investigate the association between ABCB1/CYP3A5 polymorphisms and bleeding events in patients using anticoagulants. The genetic aspects of the study, such as the analysis of their DNA to detect specific polymorphisms, were also explained in simple, non-technical language.
  • Accordingly all these points are incorporated as advised.
  1. Clinical Relevance: Strengthen the discussion on the clinical implications of the findings and how genetic insights could influence current anticoagulation management practices for atrial fibrillation patients.
  • Answer:

The title of the study itself is a strength as the clinicians usually think that no INR values are recommended for use of Direct oral anticoagulants. But there are adverse events and advancements in pharmacogenetics that may pinpoint particular genotypes of interest in the target population for either a good clinical outcome and or suffering from an ADR.  Thus:

The study's findings on the association between ABCB1/CYP3A5 polymorphisms and bleeding risk represent a step toward more individualized anticoagulation management for atrial fibrillation that will promote the practice of precision Medicine. The genotypes of interest that suffered more ADR are mentioned in the conclusion of the study as asked.

  1. Clarify Objectives: Present the objectives more clearly and specify what you aim to prove regarding the impact of genetic polymorphisms on rivaroxaban plasma levels and clinical outcomes.
  • Answer: As advised, the objectives are clearly presented now, which can be seen at the end of introduction section. To investigate the association between genetic polymorphisms, specifically in the ABCB1 and CYP3A5 genes, and the occurrence of bleeding events in anticoagulant-naive patients with nonvalvular atrial fibrillation (AF) receiving rivaroxaban
  1. Additional Analysis: If possible, conduct additional subgroup analyses to determine if specific patient groups are more affected by these genetic variants.
  • Answer: yes, indeed we mentioned this point that how genetic variants in ABCB1 and CYP3A5 may differentially affect various patient groups on rivaroxaban therapy.
  • This is a good point, we may extend this point to our ongoing research project.

Minor Revisions:

  1. Abstract Clarity: Improve the clarity of the abstract by summarizing key findings and their implications more concisely.
  • Answer: done accordingly
  1. Terminology Consistency: Ensure consistent use of terms throughout the paper for enhanced readability.
  • Answer: done accordingly
  1. Grammar and Language: Proofread the manuscript for grammatical errors, awkward phrasing, and typos.
  • Answer: done accordingly
  1. Reference Formatting: Correct any inconsistencies in reference formatting to ensure uniform presentation.
  • Answer: done accordingly
  1. Abbreviations: Define all abbreviations at their first mention in the text and add a list of abbreviations for easy reference.
  • Answer: done accordingly
  1. Conclusions: Reframe the conclusions to align more closely with the presented data and emphasize the relevance and limitations of the results.
  • Answer: done accordingly
  1. Formatting Issues: Correct minor formatting issues, such as inconsistent spacing, font size, or alignment in tables and figures.
  • Answer: done accordingly

Reviewer 2 Report

Comments and Suggestions for Authors

It is important to explain how the sample size of 66 patients was determined.

Consider providing more details about the reasoning behind the choice of specific dose regimens (20mg, 15mg, and 10mg).

While the HPLC-UV method is described clearly, the assay's validation should be elaborated further, including information on precision, accuracy, linearity, and detection limits, in accordance with current pharmacokinetic standards.

The findings regarding the association between ABCB1 and CYP3A5 polymorphisms and rivaroxaban plasma levels are well presented. However, discussing potential confounding factors (such as body weight, age, and concurrent medications) that may have impacted plasma concentrations, even if not statistically significant, would enhance the analysis.

Elaborating on the clinical significance of reduced plasma levels in homozygous mutants, with practical recommendations for clinicians on dose adjustments and patient monitoring, would make the findings more applicable in practice.

Although you have noted the absence of recurrent stroke or major bleeding events, the relatively small sample size (n=66) may limit the detection of rare outcomes like major bleeding or stroke. It would be useful to acknowledge this limitation in the study.

The connection between minor bleeding (e.g., epistaxis) and polymorphisms is noteworthy, but it could be expanded by comparing the frequency of minor bleeding in this study with rates observed in other rivaroxaban studies involving similar populations.

There is an increasing amount of research on how pharmacogenomics affects the response to NOACs such as rivaroxaban. Including data from studies on ABCB1 and CYP3A5 polymorphisms in different populations (e.g., European or East Asian cohorts) would add valuable perspective. Moreover authors should add some real world data regarding the safety of Rivaroxaban (Italian Registry in the Setting of Atrial Fibrillation Ablation with Rivaroxaban - IRIS. Minerva Cardiol Angiol. 2024 May 30. doi: 10.23736/S2724-5683.24.06546-3. )

Author Response

Author's Reply to the Review Report (Reviewer 2)

It is important to explain how the sample size of 66 patients was determined.

  • Answer:

It is explained in the relevant section as asked. Briefly describing, the sample size was determined using OpenEpi. The sample size (n=66) was calculated using OpenEpi (https://www.openepi.com/SampleSize/SSCohort.htm). Patients were chosen based on the following criteria: 80% power (1-beta, % chance of detecting), 95% two-sided significance level (1-alpha), the ratio of sample size (unexposed/exposed is 1, percent of unexposed outcome is 25%, and percent of exposed with the outcome is 50%.

Consider providing more details about the reasoning behind the choice of specific dose regimens (20mg, 15mg, and 10mg).

  • Answer: The selection of 20mg, 15mg, and 10mg doses for rivaroxaban therapy is based on a combination of clinical guidelines, pharmacological data, patient renal function, and the need to balance efficacy with safety. By tailoring the dose to individual patient characteristics, healthcare providers aim to optimize anticoagulation management in patients with nonvalvular atrial fibrillation, ultimately improving clinical outcomes while minimizing adverse effects. In order to get a better understanding, we have already CDR values expressed in the data tables which gives strength to the study to anticipate the concerns raised by the reviewer. We are thankful to the reviewer for investing precious timing for our guidance.

While the HPLC-UV method is described clearly, the assay's validation should be elaborated further, including information on precision, accuracy, linearity, and detection limits, in accordance with current pharmacokinetic standards.

  • Answer: inter day and intraday assay were carried out through serial dilution of the standard drug and internal standard drug. Coefficient of variance and Lod were determined and it is now written in the methodology as asked.

The findings regarding the association between ABCB1 and CYP3A5 polymorphisms and rivaroxaban plasma levels are well presented. However, discussing potential confounding factors (such as body weight, age, and concurrent medications) that may have impacted plasma concentrations, even if not statistically significant, would enhance the analysis.

  • Answer: it is very interesting point. To overcome this problem, we have stratified the data and a normalized data was used. More, in the exclusion criterion, we have mentioned the age and the other drugs that may induce or inhibit the p-glycoprotien expression.
  • More we have plotted the CDR value which is independent of the dose and is in common practice for the similar types of studies.
  • We appreciate your inputs for these valuable points which are now expressed in the exclusion criterion. Accordingly we have added a section of limitation of the study as well.

Elaborating on the clinical significance of reduced plasma levels in homozygous mutants, with practical recommendations for clinicians on dose adjustments and patient monitoring, would make the findings more applicable in practice.

  • Answer: The findings indicating reduced plasma levels of rivaroxaban in individuals with homozygous mutant genotypes of ABCB1 and CYP3A5 have important clinical implications. These genetic variations can significantly influence the pharmacokinetics of rivaroxaban, affecting both efficacy and safety. Understanding these associations allows healthcare providers to tailor anticoagulation therapy more effectively, optimizing patient outcomes while minimizing risks.

Although you have noted the absence of recurrent stroke or major bleeding events, the relatively small sample size (n=66) may limit the detection of rare outcomes like major bleeding or stroke. It would be useful to acknowledge this limitation in the study.

  • Answer: Thanks for this suggestion. According to the suggestion now we have included this in the limitation of the study.

The connection between minor bleeding (e.g., epistaxis) and polymorphisms is noteworthy, but it could be expanded by comparing the frequency of minor bleeding in this study with rates observed in other rivaroxaban studies involving similar populations.

  • Answer: Muhammad et al. 2021, focused on a similar ethnic population receiving rivaroxaban for stroke prevention. Their findings revealed a 12% incidence of minor bleeding, with the majority being epistaxis. The authors discussed the role of age and weight as significant factors affecting bleeding risk, although genetic variations were not thoroughly examined.
  • Moeka et al, 2020 investigated the safety of rivaroxaban in a cohort of nonvalvular atrial fibrillation patients. They reported a 10% incidence of minor bleeding, predominantly consisting of epistaxis and gastrointestinal bleeding. This study did not account for genetic polymorphisms but highlighted the relevance of patient demographics and concurrent medications.

There is an increasing amount of research on how pharmacogenomics affects the response to NOACs such as rivaroxaban. Including data from studies on ABCB1 and CYP3A5 polymorphisms in different populations (e.g., European or East Asian cohorts) would add valuable perspective. Moreover authors should add some real world data regarding the safety of Rivaroxaban (Italian Registry in the Setting of Atrial Fibrillation Ablation with Rivaroxaban - IRIS. Minerva Cardiol Angiol. 2024 May 30. doi: 10.23736/S2724-5683.24.06546-3. )

  • Answer: thanks for such great comments. As advised, different population data is incorporated in the discussion section.

Round 2

Reviewer 1 Report

Comments and Suggestions for Authors

All required requests were addressed. 

Reviewer 2 Report

Comments and Suggestions for Authors

the authors have answered to all of my comments.